# Frailty and Behavioral and Psychological Symptoms of Dementia: A Single Center Study

**DOI:** 10.3390/geriatrics9060141

**Published:** 2024-11-02

**Authors:** Sara Rogani, Valeria Calsolaro, Giulia Coppini, Bianca Lemmi, Irene Taverni, Elena Bianchi, Maria Giovanna Bianco, Rosanna Pullia, Ludovica Di Carlo, Chukwuma Okoye, Agostino Virdis, Fabio Monzani

**Affiliations:** 1Geriatrics Unit, Department of Clinical and Experimental Medicine, University of Pisa, 56124 Pisa, Italy; sara_rogani@hotmail.it (S.R.); valina82@gmail.com (V.C.); giulia.coppini1@gmail.com (G.C.); bianca.lemmi@gmail.com (B.L.); irene.taverni@gmail.com (I.T.); dr.elenabianchi@gmail.com (E.B.); mariagiovanna.bianco90@gmail.com (M.G.B.); rsn.pullia@gmail.com (R.P.); dicarlo.ludovica@gmail.com (L.D.C.); agostino.virdis@unipi.it (A.V.); fabio.monzani@unipi.it (F.M.); 2Department of Medicine and Surgery, University of Milano-Bicocca, 20126 Milano, Italy; 3Fondazione IRCCS San Gerardo dei Tintori, 20900 Monza, Italy

**Keywords:** frailty, dementia, behavioral and psychological symptoms in dementia (BPSD), hyperactivity

## Abstract

**Background:** During the time-course of cognitive decline, Behavioral and Psychological Symptoms of Dementia (BPSD) may arise, negatively impacting the outcomes. **Methods:** The aim of this single center, longitudinal study was to evaluate the correlation between frailty and BPSD in a population of older patients with dementia. BPSD were classified into three clusters: “mood/apathy” (depression, apathy, sleep disturbances, appetite disturbances), “psychosis” (delusions, hallucinations, and anxiety), and “hyperactivity” (agitation, elation, motor aberrant behavior, irritability, disinhibition). Using the Clinical Frailty Scale (CFS), patients were categorized as “severely frail”, “mild/moderately frail” and “robust” (CFS ≥ 7, 4–6, and ≤ 3, respectively). **Results:** In total, 209 patients (mean age 83.24 ± 4.98 years) with a clinical diagnosis of dementia were enrolled. BPSD were prevalent among the severely frail patients. A positive correlation at regression analysis was found between frailty and “hyperactivity” cluster at baseline and follow-up visits (*p* < 0.001, *p* = 0.022, *p* = 0.028, respectively), and was confirmed at the network analysis. Loss of independence in IADL was correlated to hyperactivity and psychosis symptoms (*p* < 0.001 and *p* = 0.013, respectively). **Conclusions:** Scarce literature is available regarding the correlation between frailty and BPSD, which in our study is significant, especially for symptoms in the hyperactivity cluster. Frailty assessment may help identify patients at the highest risk for developing BPDS who might benefit from targeted intervention in the earliest phases of the disease.

## 1. Introduction

Dementia is a clinical syndrome characterized by progressive decline in multiple cognitive domains and loss of independence in activities of daily living. Affecting 47 million people worldwide, dementia is a growing public health issue with a considerable economic impact; aging represents a major risk factor, as the prevalence of dementia almost doubles every five years after 65 years of age [1,2].

Alongside the cognitive symptoms, alterations in personality and behavioral changes, such as depression, agitation, apathy, aggression, psychosis, hallucinations, and delusions, may arise. Behavioral and Psychological Symptoms of Dementia (BPSD) may be among the earliest signs of cognitive decline [3,4]; their clinical presentation varies greatly among individuals and, although fluctuating, they become more severe over the course of the disease, causing considerable distress to both patients and caregivers [5]. BPSD are associated with several negative outcomes, such as faster cognitive decline and progression to more severe stages of dementia, loss of independence, and increased risk for secondary complications such as falls and fractures, representing the leading reason for higher hospitalization rates and early institutionalization [6].

BPSD can be grouped into distinct clusters, suggesting a possible common perturbation of signaling pathways or neural circuitry specific to each cluster. The existence of behavioral subsyndromes, with each group of symptoms reflecting a different prevalence, timeline, and biological and psychosocial factors, can help identify the possible correlations between BPSD and clinical variables, outlining specific interventions targeting BPSD subsyndromes rather than individual symptoms [7].

Considering that dementia mainly affects older people, it would be interesting to evaluate the possible link to other geriatric syndromes, and, in particular, to frailty. Frailty is defined as an aging-related syndrome, characterized by increased vulnerability to adverse events, with reduced tolerance to medical and surgical interventions [8]. Frail older patients often present with an increased burden of symptoms and are predisposed to negative health outcomes, such as falls, fractures, hospitalization, disability, poor quality of life, and with higher risk of institutionalization. Current evidence in the literature has shown a strong correlation between frailty and cognitive disorders, while the literature about the correlation between BPSD and frailty is scarce.

The aim of the current study was to evaluate the possible correlation between frailty and BPSD in a population of older patients with dementia, at baseline and over time across a year follow-up; moreover, the potential association between single BPSD and frailty was evaluated.

## 2. Materials and Methods

### 2.1. Data Collection and Participants

This is a longitudinal, retrospective, single center study enrolling patients referred to our Memory Clinic aged 65 years old and older. Inclusion criteria were reported cognitive complaint, presence of a caregiver able to corroborate or provide the clinician with medical information, and ability to attend clinical outpatient visits. Patients with diagnoses highly suspicious for frontotemporal, vascular, Parkinson’s, or Lewy Body dementia, bed-bound patients, and terminally ill patients were excluded. The study protocol complied with the Declaration of Helsinki and was approved by the local Ethic Committee (approval number 22187). Informed consent was acquired whenever possible, as per protocol for retrospective study.

Demographic characteristics and clinical history were obtained, along with a Comprehensive Geriatric Assessment (CGA) and physical and functional examination. CGA was performed by using scales exploring comorbidities, functional and cognitive performance as follows: Cumulative Illness Rating Scale (CIRS-C) [9], Basic Activities of Daily Living (BADL), Instrumental Activities of Daily Living (IADL) [10], and Mini Mental State Examination (MMSE) [11]. Frailty was assessed using the visual analog scale Clinical Frailty Scale (CFS) [12]. Patients were further categorized as “severely frail”, “mild/moderately frail” and “robust” on the basis of CFS score (CFS, respectively, ≥7, 4–6, and ≤3). The NPI scale was used in order to assess the presence and severity of neuropsychiatric symptoms; information for the NPI was obtained from the caregiver. According to Aalten et al [7], BPSD were classified into three different clusters: “mood/apathy” (depression, apathy, sleep disturbances, and appetite disturbances), “psychosis” (delusions, hallucinations, and anxiety), and “hyperactivity” (agitation, elation, motor aberrant behavior, irritability, and disinhibition) [7]. Patients were evaluated at baseline and 6 and 12 months follow-up visits; we have allowed a time window between 5 and 7 months for the first follow-up, and 11–13 months for the second visit.

### 2.2. Statistical Analysis

Statistical analysis was performed by using Statistical Package for the Social Sciences (SPSS 21.0, SPSS Inc., Chicago, IL, USA). Demographic and clinical characteristics among diagnostic groups were compared using Analysis of variance (ANOVA) for continuous normally distributed variables and the Chi-square test (χ^2^) for categorical or dichotomous variables. Continuous variables were expressed as mean ± standard deviation, ordinal variables as median and interquartile range, and categorical variables as percentage. One-way repeated measures analysis of variance (ANOVA) with Bonferroni correction was used to compare means across the follow-up visits. We used linear regression to estimate the association between the number of symptoms of all three BPSD clusters and the CFS score; logistic regression was used to analyze the relationship between categorical variables, such as the presence of one or more symptoms of a particular cluster and multiple influencing factors, including BADL, IADL, CFS, MMSE, and age. Stepwise regression was performed, considering age, CFS, BADL, IADL, MMSE, and CIRS scores as independent variables, to evaluate the possible contribution of single factors to the correlation with BPSD. Statistical significance was assigned for *p* < 0.05. We also developed a network analysis with the secondary endpoint of identifying the most influential symptom in relation to the frailty degree of our population. The nodes represent various variables, including different types of BPSD and other factors like CFS and CIRS. The edge may be positive (e.g., positive correlation) or negative (e.g., negative correlation), and the polarity of the relationship is represented graphically using different colored lines: blue for positive relationships and red for negative relationships. Varying the thickness and color density of the edges we underlined the strength of the relationships.

## 3. Results

### 3.1. Description of the Cohort

A total of 209 patients referred to our outpatient clinic from 2018 to 2021 with a clinical diagnosis of dementia (71.3% women; mean age = 83.24 ± 4.98 years) were enrolled. The main demographic and clinical characteristics of the studied subjects are summarized in Table 1.

At baseline, the study cohort showed a low burden of comorbidity (mean CIRS-C score 1.35 ± 1.5), a mild degree of frailty with a need for help in high order IADLs [CFS 5 (IQR 1), BADL 5 (IQR 3), IADL 3 (IQR 4)], and moderate cognitive decline (MMSE score 19.08 ± 5.0). The NPI scale for the assessment of BPSD showed a moderate degree of severity of symptoms and caregiver distress (NPI axb 6.63 ± 6.6, NPI distress 3.98 ± 3.6).

### 3.2. BPSD and Frailty at Follow-Up

The most represented group was the mild/moderately frail one (n = 155); lower numbers were seen in the robust (n = 18) and severely frail (n = 36) ones. The percentage of patients presenting with at least one symptom of any cluster was higher among the severely frail subgroup compared to the less frail, with 69.4% exhibiting at least one symptom of the “mood-apathy” cluster, 44.4% of the “psychosis”, and 63.9% of the “hyperactivity” clusters (Appendix A).

The repeated measure ANOVA conducted over the three timepoints showed that each pairwise difference was significant. The results of the ANOVA indicated a significant time effect on functional status, with worsening of ADL and IADL over time [ADL Wilks’ Lambda = 0.744, F(2.80) = 13.75, *p* < 0.001, η^2^ = 0.256, IADL Wilks’ Lambda = 0.71, F(2.80) = 16.3, *p* < 0.001, η^2^ = 0.29, respectively]. Likewise, there was a significant progression of cognitive impairment as measured with MMSE and a significant worsening of frailty by CFS [MMSE Wilks’ Lambda = 0.751, F(2.73) = 12.11, *p* < 0.001, η^2^ = 0.249, CFS Wilks’ Lambda = 0.749, F(2.79) = 13.224, *p* < 0.001, η^2^ = 0.241]. In particular, BADL significantly worsened between baseline and 6 months follow-up (*p* = 0.005) and between 6 months and 12 months (*p* < 0.001); IADL worsening was significant between baseline and 6 months follow-up (*p* = 0.002) and between 6 and 12 months (*p* < 0.001). The same trend was observed for the MMSE score, with a progressive significant worsening between the three timepoints (*p* = 0.041 between baseline and 6 months, *p* < 0.001, and between 6 and 12 months). The worsening of CFS was significant between baseline and 6 months follow-up (*p* < 0.001), and between 6 and 12 months (*p* = 0.025). The results of pairwise analysis are shown in Table 1.

### 3.3. Correlation Analysis Between BPSD and Functional Abilities

In light of the higher percentage of BPSD among severely frail patients, we evaluated the potential correlation between psychiatric symptoms and frailty.

At linear regression, the number of symptoms of the hyperactivity cluster correlated with the degree of frailty, at baseline and at 6 and 12 months follow-up (*p* < 0.001, *p* = 0.022, *p* = 0.028, respectively). The correlation remained significant after correction for cognitive impairment. Conversely, no association was observed with both the mood/apathy and the psychosis clusters; however, after correction for cognitive impairment, there was a correlation between the number of symptoms in the mood/apathy cluster and CFS at baseline, and with a number of symptoms on the psychosis cluster and CSF at 12 months follow-up. To evaluate the possible contribution of single factors to the correlation with BPSD clusters, a stepwise regression was performed, considering age, CFS, BADL, IADL, MMSE, and CIRS-C scores as independent variables. At baseline, the loss of independence in IADL resulted to be an independent risk factor for the “hyperactivity” and the “psychosis” clusters (*p* < 0.001 and *p* = 0.013, respectively); a minor burden of comorbidity evaluated by CIRS-C score was correlated to the “psychosis” cluster (*p* = 0.01). On the contrary, a higher burden of comorbidity and frailty and younger age concurred as significant risk factors for symptoms of the “mood/apathy” cluster, although with different strengths (*p* = 0.008, *p* = 0.04 and *p* < 0.001, respectively) (Table 2).

Logistic regression showed that at baseline, the reduced ability to perform instrumental activities of daily living (IADL) was associated with symptoms of the “hyperactivity” cluster (*p* = 0.022, OR 0.813, IC 0.68–0.97); this finding was confirmed at 6 months follow up (*p* = 0.003, OR 0.65, IC 0.49–0.87), while at 12 months a strong association emerged with CFS (*p* = 0.008, OR 2.9, IC 1.32–6.38). As for the presence of one or more symptoms of the “mood/apathy” cluster, at baseline, there was a significant positive association with CFS (*p* = 0.019, OR 1.6, IC 1.1–2.6).

In order to confirm and visually represent the complex pattern of relationships between the different factors, we performed a network analysis. In our model represented in Figure 1, not all nodes are equally important in determining the network’s structure; clustering of nodes that are highly interconnected among themselves and poorly connected with nodes outside the cluster can be identified. The degree of frailty, expressed by CFS, relates to behavioral and psychological symptoms of the hyperactivity cluster, such as agitation and motor aberrant activity; the relationship is still positive although not so strong with other symptoms like apathy and hallucinations. The clustering of nodes also reflects the clustering of different BPSD in the three subgroups we identified: psychosis, mood/apathy, and hyperactivity clusters.

## 4. Discussion

The aim of our study was to evaluate the impact of frailty on the presence, onset, and progression of specific clusters of BPSD in older patients referred to our Memory Clinic, at baseline and during follow-up. Current evidence in the literature has shown a strong correlation between frailty and cognitive disorders, including mild cognitive impairment and dementia, suggesting that cognition and frailty may interact within a cycle of age-related decline [13,14,15]. The degree of frailty could contribute to cognitive decline, with frail older adults at higher risk of developing dementia compared to robust ones [16,17,18,19].

Despite the ample literature existing on the association between frailty and dementia, to the best of our knowledge, scanty data are available on the correlation between the degree of frailty and BPSD clusters. Sugimoto et al. evaluated the association between physical frailty and BPSD in a cohort of patients with AD, with physical frailty calculated with 38 items Frailty Index (FI). The results of the study showed that the presence of physical frailty and pre-frailty increased the BPSD burden in patients with AD [20]. That is confirmed in our cohort, where the cluster of subjects with higher comorbidity burden and lower cognitive and physical performance showed a higher prevalence of psychiatric symptoms. Among our cohort of older patients, the prevalence of neuropsychiatric symptoms was higher in the “severely frail” compared to the other groups. In particular, almost two-thirds of patients of the “severely frail” subgroup showed neuropsychiatric symptoms of the “mood/apathy” cluster at baseline, remaining substantially stable during follow-up. Similar to what was observed for the “mood/apathy” cluster, the prevalence of “severely frail” patients with neuropsychiatric symptoms of the “psychosis” and “hyperactivity” clusters was always higher compared to patients of the “mild/moderate frail” subgroup, suggesting that the degree of frailty could be associated with the burden of BPSD symptoms. Accordingly, as shown in the network analysis graph, the degree of frailty at baseline is positively correlated to agitation and motor aberrant activity, both belonging to the hyperactivity cluster. However, the contributing role of BPSD to frailty and, possibly, the reverse is yet to be determined. Consistently, regression analysis confirmed that reduced score in the IADL appears to be correlated with symptoms of the hyperactivity and psychosis cluster clusters, which may suggest either that the loss in higher functions could be a trigger for aberrant behaviors in subjects with cognitive impairment, or that the presence of hyperactivity or psychosis symptoms may affect the ability of efficiently performing complex activities of daily living. A higher degree of frailty was also associated with symptoms of the mood apathy cluster at baseline, suggestive of depression across frailer subjects. The correlation of BPSD with the burden of comorbidity was inhomogeneous across different clusters, and the actual significance of it would need larger cohorts to be further clarified.

Although the literature about the possible correlation between BPSD and frailty is scarce, a lot has been written about the association between frailty and delirium. Postoperative Cognitive Dysfunction (POCD) and delirium (POD) are the most common perioperative cognitive complications in older patients undergoing different surgical procedures, such as spine surgery [21] or major noncardiac surgery [22]. A meta-analysis by Zhang et al. of a total of 30 independent studies from nine countries, consisting of 217,623 hospitalized patients (medical, surgical, emergency, and critical illness patients), showed an increased risk of delirium in frail patients compared to those not frail [23].

Among the different BPSD clusters, hyperactivity and psychosis share some features with delirium, in particular with hyperkinetic and mixed delirium. Considering that delirium usually presents with symptoms that are common to the “hyperactivity” cluster, the association between this cluster and frailty is not a surprise. Moreover, it should be considered the possibility that some diagnosis of delirium, especially if and when lasting for a long time after discharge, could rather be a symptom of BPSD and a sign of dementia, supporting the need for cognitive follow-up of all patients developing in-hospital delirium.

The results of our research also suggest the usefulness of an in-depth study in a larger cohort of patients with cognitive impairment and BPSD. A strength of our study is that the cohort of patients could be considered representative of the general oldest population, since our sample is composed of patients referred from the community for cognitive complaints. Even though the results of the study are promising and interesting, we have to acknowledge a few limitations. First of all, the patients enrolled have a clinical diagnosis of Alzheimer’s disease or mixed dementia, based on clinical criteria and structural imaging, but without the biomarkers, we could not exclude some misdiagnoses. However, patients with highly suspicious frontotemporal dementia, Lewy body dementia, Parkinson’s disease dementia, or post-stroke dementia were not included in the dataset. It also should be acknowledged that this study has been conducted in a time window comprehensive of the pandemic crisis, when social interactions were forbidden, or at least limited. Thus, physical or behavioral worsening might partially be a consequence of the social situation, and studies on a larger scale would be desirable.

Moreover, the patients presenting BPSD were a small number, and a wider cohort would have been more informative, especially enrolling more patients “robust” and “severely frail”. A further analysis evaluating not just the single cluster, but also the combination of them would certainly be more informative and representative of the general population; however, the present cohort is not sufficiently large for further division into subgroups.

## 5. Conclusions

The purpose of the current study was to determine if the presence of physical frailty, as determined by the CFS, could represent a risk factor for the development and progression of BPSD in older patients with AD. One of the most significant findings that emerged from this study is the association between frailty and reduced independence with symptoms of the “hyperactivity” and “psychosis” clusters. However, whether the loss of independence is a possible cause of frailty or vice versa is still to be determined. The results of this study suggest that the assessment of frailty may help identify patients at risk of developing behavioral and psychological symptoms of dementia, and could provide clinicians with a time window to target intervention in the earliest phases of both frail condition and BPSD. This is of crucial importance, especially because early intervention may benefit from non-pharmacological approaches, saving patients from the prescription of antipsychotic drugs which are linked to several adverse events. Given the importance of frailty assessment, especially in the oldest population, a standardized approach to physical frailty evaluation in future clinical studies is highly desirable and would provide clinicians with simple and highly efficient tools for the patients’ care. Further larger studies are needed to better establish the complex crosstalk between BPSD and frailty.

## Figures and Tables

**Figure 1 geriatrics-09-00141-f001:**
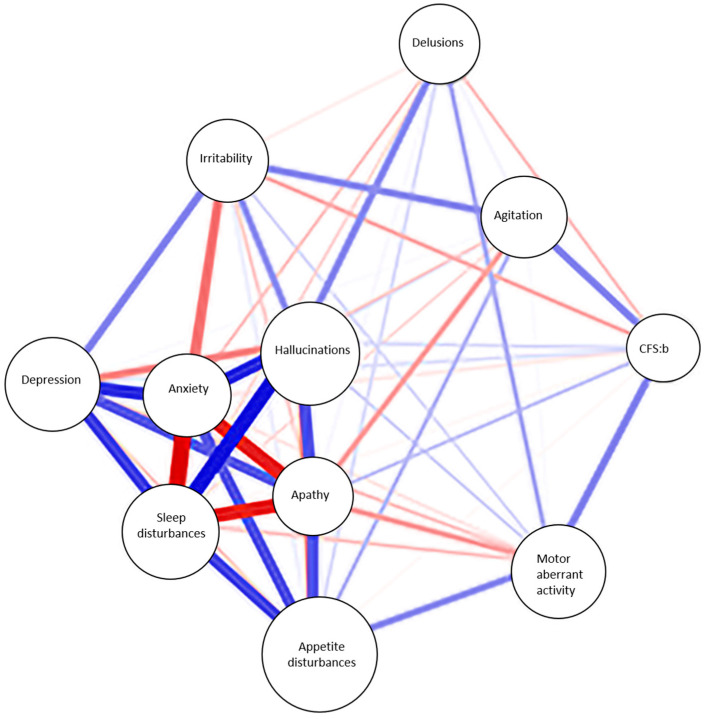
Network analysis of the correlation between CFS and neuropsychiatric symptoms. **Abbreviations:** CFS:b: clinical frailty scale at baseline. Blue lines represent positive relationships and red lines represent negative relationships.

**Table 1 geriatrics-09-00141-t001:** Characteristics of the study population.

	Baseline 209	6 Months Follow-Up 146	12 Months Follow-Up 98
Age (mean ± SD)	83.24 ± 4.99		
Gender F (%)	71.3		
CIRS score (mean ± SD)	9.26 ± 5.4		
CIRS-C (mean ± SD)	1.35 ± 1.5		
CFS (median, IQR)	5 (1) ^*,#^	6 (1.25)	6 (1.5)
BADL (median, IQR)	5 (3) ^*,#,^^	4 (3.25)	4 (3)
IADL (median, IQR)	3 (4) ^*,#,^^	2 (4)	1 (3)
MMSE (mean ± SD)	19.08 ± 5.0 ^*,#,^^	17.71 ± 5.8	17.02 ± 5.8
NPI (frequency × severity) (mean ± SD)	6.63 ± 6.6	7.53 ± 7.4	7.85 ± 7.7
NPI distress (mean ± SD)	3.98 ± 3.6	4.83 ± 4.4	4.92 ± 4.3

Repeated measures ANOVA: * *p* < 0.05 between b and 6 months. # *p* < 0.05 between b and 12 months. ^ *p* < 0.05 between 6 months and 12 months. Abbreviations. CIRS-C: Cumulative Illness Rating Scale-comorbidity; CFS: clinical frailty scale; BADL: basic activities of daily living; IADL: instrumental activities of daily living; MMSE: mini mental state examination; NPI: neuropsychiatric inventory; IQR: interquartile range.

**Table 2 geriatrics-09-00141-t002:** Stepwise regression analysis of the correlation between functional and cognitive measures with BPSD clusters.

	AGE (β Coeff Value)	BADL (β Coeff Value)	IADL (β Coeff Value)	CFS (β Coeff Value)	CIRS-C (β Coeff Value)	MMSE (β Coeff Value)
Hyperactivity	NS	NS	0.001 (−0.06)	NS	NS	NS
Mood/apathy	<0.001 (−0.18)	NS	NS	0.04 (0.18)	0.008 (0.18)	NS
Psychosis	NS	NS	0.013 (−0.17)	NS	0.016 (−0.16)	NS

Abbreviations. NS: not significant; BADL: activities of daily living; IADL: instrumental activities of daily living; CFS: clinical frailty scale; CIRS-C: Cumulative Illness Rating Scale-comorbidity; MMSE: mini mental state examination.

## Data Availability

The datasets generated during and/or analyzed during the current study are available from the corresponding authors upon reasonable request.

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
