# Peer review of "Frailty and Behavioral and Psychological Symptoms of Dementia: A Single Center Study"

_geriatrics, 2024, doi:10.3390/geriatrics9060141_

Round 1

Reviewer 1 Report

Comments and Suggestions for Authors

The authors investigated the association between frailty and psychological symptoms of dementia (BPSD). Most research, so far, has been based on the association of frailty with cognitive symptoms. Frailty is measured with a visual analogue scale; then categorized into “robust”, “mild/moderately frail” and “severely frail”. BPSD were categorized into “mood/apathy”, “psychosis” and “hyperactivity”. Patients were evaluated at baseline and 6 and 12 months later. The longitudinal aspect is a strong asset of the study.

The supplementary tables show an association of the stages of frailty to mood/apathy, psychotic symptoms and hyperactivity. What is the extra finding of the network analysis (see Figure 1)? Here we have a correlation of frailty (CFS-b) with agitation and motor aberrant activity. And we have a clustering of the BPSD-symptoms; no new results, except a validation of the BPSD syndrome.

The presentation of the statistical results is very confusing. Section 3.3. presents results of longitudinal regressions of hyperactivity on BPSD; no results with BPSD and mood/apathy and psychosis. This is contrary to the analysis of variance results in section 3.2 where more frailty were connected to more BPSD and all clusters. Subsequently all BPSD clusters were regressed on the variables CFS, ADL, IADL, MMSE and CIRS. There were scattered significant results: IADL at baseline only with “hyperactivity”, CIRS-C (now with upper case letter C) only with “hyperactivity” and frailty and age only with “mood/apathy”. The authors should invest more time in presenting the results more consistently and clearly, maybe with a table.

It would be interesting to see the association between frailty and BPSD controlling for cognitive impairment. Then one would have an estimate of this association beyond cognitive functioning.

Minor issues:

-          What is CIRS-s? CIRS-c is burden of comorbitity.

-          Page 4, line 164: there is a word missing “resulted …. the independent risk factor”

-          -Page 1, line 45: to my understanding, there is an association, but no strong association between BPSD and cognitive decline. Please elaborate the statement, that “severity exponentially increases over the course of the disease”.

Comments on the Quality of English Language

Word are missing in a sentence (see comments to authors).

Author Response

The authors investigated the association between frailty and psychological symptoms of dementia (BPSD). Most research, so far, has been based on the association of frailty with cognitive symptoms. Frailty is measured with a visual analogue scale; then categorized into “robust”, “mild/moderately frail” and “severely frail”. BPSD were categorized into “mood/apathy”, “psychosis” and “hyperactivity”. Patients were evaluated at baseline and 6 and 12 months later. The longitudinal aspect is a strong asset of the study.

COMMENT 1: The supplementary tables show an association of the stages of frailty to mood/apathy, psychotic symptoms and hyperactivity. What is the extra finding of the network analysis (see Figure 1)? Here we have a correlation of frailty (CFS-b) with agitation and motor aberrant activity. And we have a clustering of the BPSD-symptoms; no new results, except a validation of the BPSD syndrome.

RESPONSE 1: we agree with the Reviewer that the Network analysis is a confirmation of the results presented in the supplementary tables. We have amended the manuscript stating that this analysis was run as a confirmatory one aiming at a visual representation of the results. If the Reviewer would prefer, we would remove it (line 201 of the tracked version of the manuscript)

COMMENT 2: The presentation of the statistical results is very confusing. Section 3.3. presents results of longitudinal regressions of hyperactivity on BPSD; no results with BPSD and mood/apathy and psychosis. This is contrary to the analysis of variance results in section 3.2 where more frailty were connected to more BPSD and all clusters.

Subsequently all BPSD clusters were regressed on the variables CFS, ADL, IADL, MMSE and CIRS. There were scattered significant results: IADL at baseline only with “hyperactivity”, CIRS-C (now with upper case letter C) only with “hyperactivity” and frailty and age only with “mood/apathy”. The authors should invest more time in presenting the results more consistently and clearly, maybe with a table.

 RESPONSE 2: we acknowledge that the result section was confusing. We have created a table to display the linear stepwise regression (Table 2) and we have simplified the comments. The regression was run to clarify whether the worsening of CSF and BPDS would be correlated, due to their mutual change over time as shown in the ANOVA. We hope that the updater Results session (lines 123-213 of the revised manuscript) would meet the Reviewer’s standard

COMMENT 3: It would be interesting to see the association between frailty and BPSD controlling for cognitive impairment. Then one would have an estimate of this association beyond cognitive functioning.

RESPONSE 3: we thank the Reviewer for the suggestion; we have run correction for MMSE score, and the correlation with the hyperactivity cluster remained significant across the three timepoints. A correlation with the number of symptoms in the mood/apathy cluster and CFS at baseline was noted after correction for MMSE, in line with the results from the logistic regression. Significant correlation was found at 12 months follow up in the psychosis cluster.  The results from regression were in line with the worsening resulted from repeated measures anova. The discussion of the results was updated accordingly in the manuscript

Minor issues:

-          What is CIRS-s? CIRS-c is burden of comorbidity.

RESPONSE: we are sorry for the misleading index; CIRS-S represents the severity of the burden of comorbidity. We have kept the CIRS-C in the Manuscript, which represent the burden of comorbidities, and removed the CIRS-, to avoid redundancies.

-          Page 4, line 164: there is a word missing “resulted …. the independent risk factor”

RESPONSE: we apologize, we have corrected the typos

-          Page 1, line 45: to my understanding, there is an association, but no strong association between BPSD and cognitive decline. Please elaborate the statement, that “severity exponentially increases over the course of the disease”.

RESPONSE: we are sorry for the unclear sentence, we have amended it

Reviewer 2 Report

Comments and Suggestions for Authors

Rogani et al. studied frailty and behavioral and psychological symptoms of dementia.

In the abstract, specify what statistical significance is applied. “A positive correlation between frailty and “hyperactivity” cluster was found at baseline and follow-up visits (p<0.001, p=0.022, p=0.028 respectively)”

Methodology

Please describe the inclusion and exclusion criteria

Did the authors hold the right to use those scales?

Please explain why these scales were performed.

Please explain what data was obtained from the patients.

At first time, all abbreviations should be fully described. “SPSS.”

Please provide a table with the ANOVA results.

Please review your references. They are not according to the “Instruction for authors.”

Please review the manuscript thoroughly; there are some grammatical English errors.

Author Response

Rogani et al. studied frailty and behavioral and psychological symptoms of dementia.

COMMENT 1: In the abstract, specify what statistical significance is applied. “A positive correlation between frailty and “hyperactivity” cluster was found at baseline and follow-up visits (p<0.001, p=0.022, p=0.028 respectively)”

 RESPONSE: we thank the Reviewer for the suggestion, we have amended as recommended (line 26 of the revised version of the Manuscript)

Methodology

COMMENT 2: Please describe the inclusion and exclusion criteria

RESPONSE: We are sorry for the lack of information. have added inclusion and exclusion criteria

COMMENT 3: Did the authors hold the right to use those scales?

RESPONSE: apologies if we did not specify it, the scales are available for the use in clinical work

COMMENT 4: Please explain why these scales were performed.

RESPONSE: we apologize for not being more clear in the text; we have decided to use scales globally known and part of usual geriatric assessment in clinical practice

COMMENT 5: Please explain what data was obtained from the patients.

RESPONSE: apologies if the data were unclear, we have specified in the methods which data are explored with the scales

COMMENT 6: At first time, all abbreviations should be fully described. “SPSS.”

RESPONSE: we are sorry for missing it. We have specified the full acronym

COMMENT 7: Please provide a table with the ANOVA results.

RESPONSE: we apologize for the lack of clarity in Table 1; the statistics results reported are from the Repeated measure ANOVA (we have added into the table description)

COMMENT 8: Please review your references. They are not according to the “Instruction for authors.”

Please review the manuscript thoroughly; there are some grammatical English errors.

RESPONSE: we thank the Reviewer and we apologize for the mistake. We have amended the Reference list according to the instruction for Authors and we have revised the manuscript thoroughly.

Round 2

Reviewer 1 Report

Comments and Suggestions for Authors

All points of the reviews were addressed adequately.

Reviewer 2 Report

Comments and Suggestions for Authors

Satisfactory